# Self-Assembly of Magnetic Nanoparticles in Ferrofluids on Different Templates Investigated by Neutron Reflectometry

**DOI:** 10.3390/nano10061231

**Published:** 2020-06-24

**Authors:** Katharina Theis-Bröhl, Apurve Saini, Max Wolff, Joseph A. Dura, Brian B. Maranville, Julie A. Borchers

**Affiliations:** 1University of Applied Sciences Bremerhaven, An der Karlstadt 8, 27568 Bremerhaven, Germany; 2Department for Physics and Astronomy, Uppsala University, Lägerhyddsvägen 1, 752 37 Uppsala, Sweden; apurve.saini@physics.uu.se (A.S.); max.wolff@physics.uu.se (M.W.); 3NIST Center for Neutron Research, 100 Bureau Drive, Gaithersburg, MD 20899-6102, USA; joseph.dura@nist.gov (J.A.D.); brian.maranville@nist.gov (B.B.M.); julie.borchers@nist.gov (J.A.B.)

**Keywords:** magnetic nanoparticles, ferrofluid, core/shell nanoparticles, polarized neutron reflectivity, surfactant coating, self-assembly process, surface wetting

## Abstract

In this article we review the process by which magnetite nanoparticles self-assemble onto solid surfaces. The focus is on neutron reflectometry studies providing information on the density and magnetization depth profiles of buried interfaces. Specific attention is given to the near-interface "wetting" layer and to examples of magnetite nanoparticles on a hydrophilic silicon crystal, one coated with (3-Aminopropyl)triethoxysilane, and finally, one with a magnetic film with out-of-plane magnetization.

## 1. Introduction

Self-assembly is the key route to well-ordered nanoparticle (NP) structures. It is an economical method which delivers a straightforward, controllable mechanism for arranging NPs into three-dimensional ordered structures. The rich, diverse phenomena observed during self-assembly are mainly driven by steric, electrostatic and/or magnetic interactions [1,2,3,4]. Deep knowledge of the underlying principles during the self-assembly of such systems is crucial for engineering smart, functional or stimuli-responsive synthetic composites. Self-assembled nanostructures also show remarkable collective properties [5] useful for designing nanoarchitectures, and the ordering of nanoparticles into multidimensional nanocrystalline structures may even lead to new properties compared to isolated single nanoparticles [6]. For the construction of these nanoparticle systems, the interaction of the first layer (which is subsequently referred to as the "wetting" layer to convey its nearly uniform lateral coverage) with a substrate is of great importance, since it forms the seed layer for structures grown above.

Magnetic NPs are particularly intriguing as a result of their applications in magnetic storage technologies and biomedicine [7,8,9,10,11,12,13,14]. In particular, Fe${}_{3}$O${}_{4}$ NPs have low toxicity, robust structures and strong responses to magnetic fields, allowing applications such as superparamagnetic relaxometry [15,16,17], magnetic particle imaging [18,19,20] and hyperthermia therapy [21,22]. To control their collective interactions and colloidal stability, Fe${}_{3}$O${}_{4}$ and related magnetic NPs are typically coated with functional, organic layers, allowing their suspension in a wide range of solvents, including water. In these suspensions, magnetic NPs can organize into local structures, including clusters, linear chains, branched chains and rings [23,24,25]. If no magnetic field is present, the formation process is primarily guided by magnetic dipolar interactions [26] that are significant when individual NPs consist of a single magnetic domain [4,23,25,27,28,29,30]. Besides understanding the collective interaction of solvated magnetic NPs, their self-assembly on surfaces and the influence of external magnetic fields are of interest from a fundamental perspective and for applications.

Laboratory techniques such as electron microscopy are used for studying the ordering of the magnetic nanoparticles on substrates [31,32,33,34,35,36]. The thickness or layering of particle films can be extracted from ellipsometry [34] or X-ray characterization methods [36,37]. Magnetization measurements from bulk magnetometry or micro-/spectroscopic methods provide valuable information related to the collective response of spin-textures in self-assembled nanoparticle structures [29,33,37,38]. Additionally, magnetic force microscopy is frequently used for determining the properties of magnetic NPs assembled on a surface [29,36,39]. In recent years, neutron reflectometry and small angle neutron scattering (SANS) have proven to be indispensable methods for characterizing the spin configurations of these nanoparticle systems in detail, with nanometer-scale resolution [40,41,42,43,44,45,46,47,48].

In magnetic nanoparticle assemblies, the high penetration power and isotope contrast variation further provide a basis for understanding the inter and intra-particle structures and the interactions with a surface. In this paper we review neutron reflectometry (NR) investigations of the self-assembly of magnetic nanoparticles on solid substrates with different surface conditions that could be optimized for specific technological applications. We demonstrate how density and magnetization depth profiles are extracted from specular NR and/or polarized neutron reflectivity (PNR) measurements. Particular emphasis is given to three studies of iron oxide nanoparticles that highlight the formation of wetting layers at a hydrophilic silicon crystal terminated with (1) the native oxide SiO${}_{2}$ [49], (2) a layer of (3-Aminopropyl)triethoxysilane (APTES) [50] and (3) a ferrimagnetic layer that forms magnetic domains due to its out-of-plane anisotropy [51,52].

## 2. Neutron Scattering for Studying Self-Assembly on Solid Surfaces

One of the compelling attributes of neutron scattering is that the nuclear and magnetic scattering lengths are well established (with small uncertainties) for all bulk elements and isotopes. This property enables precise determination of the distribution of structural and magnetization densities on an absolute scale. In general, the interaction between the intrinsic dipole moment of the neutron and the unpaired electron spins in the sample provides unparalleled sensitivity to magnetic structure. While neutron scattering is widely recognized as a unique probe of ensemble-averaged magnetic correlations and excitations in bulk materials, its depth and interface sensitivities to both the nuclear and magnetic potentials distinguish it from other techniques in studies of magnetic thin films and lateral nanostructures [40,41,42,43,44], especially with the additional application of polarization techniques [45,46].

Depth-dependent profiles of the nuclear structure and the magnetization, for a ferrofluid (FF) in contact with a solid surface, can similarly be extracted in situ from NR and PNR measurements. The typical scattering geometry for neutron reflectivity studies with FFs is schematically shown in Figure 1. Note that the incident neutrons traverse through a thick silicon wafer that also serves as an outer wall of the sample wet cell. To further reduce the background and to enhance the scattering contrast, H${}_{2}$O in the FF is usually exchanged with D${}_{2}$O. If directed under a glancing angle onto an interface, neutrons are transmitted, specularly reflected or scattered off-specularly by the in-plane features in the nuclear and magnetic structures. As an addition, polarized capabilities are particularly powerful for determining the depth-dependent vector magnetization of the NP assembly as the result of the strong dependence of the scattering contrast on the orientation of the neutron polarization relative to the magnetic induction [53].

For polarized neutrons (Figure 1), an Fe/Si supermirror and an Al-coil spin flipper are routinely used to select the spin state of the incident neutron beam either parallel (+) or antiparallel (-) to the applied field at the sample position. A second supermirror and flipper assembly are positioned after the sample to analyze the spin state of the scattered beam (+ or -). Though polarization efficiencies are typically higher than 97%, the data are corrected for the polarization efficiency and instrument background. While the neutron reflectivity includes contributions from both the structural and magnetic order in the sample, this scattering results only from the projection of the sample magnetization that is perpendicular to the scattering wavevector $q_{z}$ (and thus perpendicular to the sample plane) in accordance with neutron selection rules [45]. In PNR measurements, the non-spin-flip (NSF) intensities, $I_{\left(++\right)}$ and $I_{\left(--\right)}$, are sensitive to the nuclear scattering length density (SLD) depth profile Nb(z), in which *N* is the number density and *b* is the bound coherent nuclear scattering length characteristic of the material. The difference between the NSF intensities is related to the in-plane component of the magnetic SLD $Np_{∥}\left(z\right)$ that is parallel to the applied guide field and to the spin quantization axis of the neutron beam. (Note that the magnitude of p is proportional to the sample magnetization.) In contrast, the spin-flip (SF) intensities, $I_{\left(+-\right)}$ and $I_{\left(-+\right)}$, are sensitive to the projection of the in-plane magnetic SLD $Np_{⊥}\left(z\right)$ that is perpendicular to the guide field. For the study of magnetic NPs on solid surfaces, any SF scattering from the in-plane, perpendicular magnetization typically averages to zero within the coherence volume of the neutron beam.

To determine the depth dependence of the nuclear and magnetic SLD profiles, the reflectivity data are fit to a slab model for the SLDs based upon the Parratt formalism [54], which is implemented, for example, in the Refl1D software package [55,56] using the super-iterative algorithm [57]. In general, the nuclear SLD profile represents the depth-dependent composition of the sample averaged across the sample plane. For nanoparticles in solution ordering at a surface, the individual layers consist of a mixture of NP core material, NP shell material (from the surfactants) and solvent. The resulting SLDs for each layer in the profile are the in-plane average of these values weighted with the respective volume fraction and integrated over the coherence volume of the neutron beam [58]. Therefore, the determination of the local structure of the nanoparticle assembly relies on a comparison of the fitted SLD values for each layer to those calculated from models that assume a certain packing fraction. Furthermore, it is also possible to extract magnetic SLDs, which are related to the depth-dependent vector magnetization averaged across the sample plane, by fitting PNR in order to pinpoint simultaneous evolution of the nuclear and magnetic structure induced by an applied magnetic field, for example.

Here we briefly summarize recent results obtained using NR and PNR to probe the self-assembly of magnetic NP systems in close proximity to solid substrates. Following that discussion, we describe in greater detail the effects of chemical termination and applied magnetic field on the self-assembly process of magnetite NPs on contrasting silicon surfaces. In an early investigation, Avdeev et al. [59] employed NR for studying the adsorption of surfactant-coated magnetic NPs from stable FFs with different solvents (benzene and D${}_{2}$O) on functionalized silicon with a native oxide layer. In a further study, Gapon et al. [60] used NR for exploring the adsorption of nanoparticles from water-based FFs on a silicon surface. Two kinds of FFs were considered: (1) FFs with magnetite NPs coated by a double layer of sodium oleate, and (2) FFs with cobalt ferrite NPs stabilized by lauric acid/sodium n-dodecylsulphate. The authors reported the formation of a single self-assembled layer for both FFs. Furthermore, Kubovcikova et al. [61] studied the adsorption of NPs from magnetic fluids onto silicon surfaces (with a native oxide) by NR and related it to the bulk structure extracted from SANS.

The dependence of the FF ordering at SiO${}_{2}$ interfaces on magnetic field was first studied systematically by Vorobiev et al. [62] using in situ NR. The authors reported the formation of smectic-like phases in a FF in contact with a SiO${}_{2}$ surface. Up to 30 ordered layers of magnetite NPs form in the presence of small magnetic fields applied perpendicular to the sample plane, and short-ranged ordered structures appear in magnetic fields applied parallel to the plane. The authors also described the formation of a wetting double-layer, which serves as the template for the ordered NP slabs.

Theis-Bröhl et al. [53] used a combination of PNR and grazing incidence small-angle neutron scattering (GISANS) for deducing the structural and magnetic characteristics of dropcast and self-assembled cobalt-oleyl amine nanocomplexes. The authors found that the NPs self-organize into a three-dimensional hexagonal lattice with well-defined positional order over a few interparticle spacings. In later studies, PNR, GISANS and grazing incidence small-angle x-ray scattering (GISAXS) were employed to determine the structural and magnetic correlations of dropcast cobalt and spin-coated magnetite NPs on silicon substrates [63]. Relative to the hexagonal self-ordering observed for iron oxide NPs, the dropcast cobalt NPs showed a much lower degree of self organization. Mishra et al. [58] spin-coated iron oxide NPs onto a bare silicon wafer (with native oxide) and onto a vanadium film sputtered on sapphire. PNR measurements revealed the presence of a dense monolayer of NPs along with the formation of quasi-magnetic domain-like configurations.

A magnetic polymer nanocomposite in contact with a hydrophilic or hydrophobic silicon interface was studied by NR and off-specular scattering by Saini and Wolff [64]. They reported that small quantities of magnetic micelles can facilitate the crystallization of Pluronic F127 micelles solvated in water into single crystalline structures via a micro-shear effect under applied magnetic field. Later, Wolff et al. [65] combined polarized and time-resolved GISANS measurements on similar samples mixed with magnetic NPs. They showed that NR, off-specular scattering and GISANS are, in principle, able to track the self-assembly process on time scales below one second and to also provide information on the magnetic induction in the sample.

Lauter-Pasyuk et al. [66] applied specular reflection and off-specular neutron scattering to determine the distribution of magnetite NPs in symmetric block-copolymer films. The authors determined the distribution of the NPs within the lamellae and observed a distortion of the lamellar order of the copolymer matrix due to the presence of the magnetic NPs. In a related study, the authors investigated parallel and perpendicular lamellar phases in copolymer NP multilayer structures. They found that for specifically-prepared substrates, the orientation of the lamellar structure changed from perpendicular to parallel to the surface after the incorporation of magnetic NPs into the copolymer matrix. In addition, the study describes the morphologies of thin, self-assembled nanocomposite films comprised of an asymmetric diblock copolymer with embedded NPs [67].

Collectively these previous studies illustrate the power of neutron scattering for characterizing the self-assembly of NPs from colloids in proximity to different solid surfaces. Detailed understanding of the magnetic and surface interactions driving the formation of these varied structures, however, clearly requires precise control of the surfaces of both the nanoparticles and the solid substrate. Successful approaches for preparing and combining the NP and substrate components are detailed below, as determined from the three studies of iron oxide nanoparticles on different surfaces that are the focus of this review [49,50,51,52].

## 3. Nanoparticle Surface Coating

Iron oxide NPs are typically synthesized by the thermal decomposition method (organometallic route), yielding particles highly uniform in all respects (size, shape, composition and crystallography) [68,69,70]. Generally, oleic acid is used as the surfactant. It is an amphiphilic molecule with a hydrophobic methyl end and hydrophilic carboxylic head, and it forms a dense protective monolayer on the particle surface that prevents agglomeration through steric repulsion (Figure 2a). Unfortunately, these surfactant coatings on the NPs render them insoluble in polar solvents, like water, which can be crucial for biological applications. Therefore, highly monodispersed magnetite NPs are generally prepared in organic solvents and thereafter phase transferred to water. The process involves addition of high oleic acid concentrations (0.6–0.8 wt.% to iron oxide) for ex-situ modifications to the as-synthesized NPs dispersed in a polar carrier with an adjusted pH, generating double layer-oleic acid modified particles [71,72]. In such particles, the primary layer is covalently linked to the NP surface through coordinate bonding between the NP and the carboxyl groups of the oleic acid, while the secondary layer is physically absorbed on the primary layer by forming an interpenetration layer with the tails of the primary layer (Figure 2b).

In another approach, as-synthesized NPs in an organic solvent are functionalized with an amphiphilic polymer such as poly(maleic anhydride-alt-1-octadecene)-poly(ethylene glycol) (PMAO-PEG) [Figure 2c], which adsorbs onto the NP surface [30,74,75]. The obtained water-dispersible NPs possess a hydrophobic inner shell and a hydrophilic corona (Figure 2d). These surface modifications offer hydrophilic terminations, such as carboxyl acids (-COOH), that are capable of covalent conjugation with appropriate functional groups (e.g., amines, -NH${}_{3}$) on solid surfaces [50].

## 4. Substrate Templates

The substrate choice and its surface preparation are also critical for determining the characteristics of the resulting self-assembled NP structures, as exemplified by our three representative investigations. In the first approach [49] a hydrophilic silicon oxide surface (cleaned with ethanol and nitrogen gas) can be used as a template for the magnetite NPs from the FF to wet the surface. To produce a more hydrophilic surface, the silicon can be etched by pirhana solution which results in a contact angle for water of zero degrees. Since oleic acid is very soluble in water, it can be assumed that the magnetic NPs wet the hydrophilic Si surface.

Alternatively, (3-aminopropyl)triethoxysilane (APTES) can be coated onto a silicon wafer to provide a template for wetting nanoparticles when the FF is in contact [50]. The APTES layer introduces a positive partial charge on the surface that provides anchors for the negatively-charged carboxylic groups of the NP coating (Figure 2b). This configuration results in an even stronger affinity between the magnetic NPs and the silicon surface. The interaction between the carboxylic groups in the NP shells and the silane groups in the APTES layer is depicted in Figure 3.

In a third approach [51,52] the template is provided by a magnetic surface with out-of-plane anisotropy, such as Tb${}_{15}$Co${}_{85}$ films [51,52]. Figure 4 shows magnetic force microscopy (MFM) images [52] of the magnetic domain structure for this system, as an example. It is expected that the NPs move along the field gradient generated by the out-of-plane stray fields and stick to the surface.

## 5. Self-Assembly of Nanoparticles on Solid Surfaces

Building upon in-depth knowledge of the structural characteristics of the iron oxide NP components and the prepared silicon surfaces, we describe the specular NR and PNR measurements of the self-assembly process from solution and its dependence upon particle size and symmetry, substrate surface termination, and magnetic field [49,50,51,52]. Particularly, we highlight the specific conditions under which densely packed hexagonal layers can be formed at the silicon-ferrofluid interface.

### 5.1. Self-Assembly of Nanoparticles at a Hydrophilic Silicon Surface

A water-based FF from Liquids Research Limited with an approximate particle diameter of 10 nm and an oleic acid-based ligand dispersant providing a shell thickness of 1–3 nm, was diluted to produce a mixture of 70.5 vol% D${}_{2}$O and 4.6 vol% of iron oxide NPs with residual ligands. (Certain commercial equipment, instruments, materials, suppliers and software are identified in this paper to foster understanding. Such identification does not imply recommendation or endorsement by the National Institute of Standards and Technology, nor does it imply that the materials or equipment identified are necessarily the best available for the purpose.) Additional sample details are provided in a prior publication [49]. Figure 5a depicts the neutron reflectivity plotted versus $q_{z}$ for the FF abutting a Si/SiO${}_{2}$ surface in magnetic fields of 0 mT and 11 mT. (Throughout the manuscript error bars represent one standard error.)

The SLD profiles extracted from fits to the data (solid lines) are shown in Figure 5b,c, respectively. For comparison SLD values for the bare magnetite core, water mixture and oleic acid shell material, and the calculated value of the core/shell nanoparticle are included as gray dashed lines.

Following the silicon oxide base, the SLD profile in 0 mT is explained by a first layer (# 1) that consists of oleic acid associated with the ligand shell surrounding the NPs. The second layer (# 2) is composed primarily of magnetite cores laterally separated by oleic acid and water. Despite nonuniformities among the NP shapes, this layer is densely packed as revealed by comparisons to SLD model calculations shown by the blue dashed line in Figure 5b. The next layer (# 3) consists mainly of ligands, similar to layer # 1. Schematics of the top and side views of the six-fold arrangement of close-packed spherical particles are drawn in Figure 6. The loosely-packed transition layer (# 4) that forms above the wetting layer is illustrated (along with the other layers) in Figure 7. Note that all layers also contain varying amounts of D${}_{2}$O/H${}_{2}$O.

When an 11 mT field is applied in-plane, the resultant SLD value for layer (# 2) is higher than in 0 mT. The SLD value derived from the model calculations performed for close-packed spherical particles can no longer describe the experimental data. Instead, rotation of the nanoparticles in the magnetic field increases the packing density and subsequently reduces the effective core and shell thicknesses [49]. Specifically, the shape anisotropy generates a force that twists the elliptical particles and reorients them with their major axis parallel to the sample surface, as illustrated in the magnetic field model in Figure 7.

### 5.2. Self-Assembly of Nanoparticles on an APTES Coated Silicon Surface

Dense, well-defined NP layers were reported to form on top of a (3-aminopropyl)triethoxysilane (APTES) coating on a silicon wafer [50]. The dilute FF mixture (i.e., nominal concentration of 0.15 vol% Fe${}_{3}$O${}_{4}$) used for this investigation contained monocrystalline, spherical magnetite NPs with a core diameter of 25 nm [30] coated with a monolayer of oleic acid and a monolayer of an amphiphilic polymer with carboxylic acid. Since the NPs are ferromagnetic at room temperature, they form stable dimer and trimer chains due to dipolar interactions, as revealed by SANS measurements [50]. Figure 8a shows the PNR as a function of $q_{z}$ for this sample in magnetic fields of 6 mT and 100 mT. The nuclear SLD profiles extracted from fits to the data (solid lines) are plotted in Figure 8b,c, respectively.

The SLD profiles can be interpreted by assuming a densely-packed NP wetting layer (# 1–3), which is effectively segregated into ligand, NP core and less-dense ligand sublayers, as schematically shown in Figure 9a. Only a small amount of water resides in the close-packed regions, and the spaces between the NPs are filled with surplus ligand material. It appears that the amide bond between the amine group of the APTES and the carboxylic groups of the shell [76] tightly bind the NP wetting layer onto the APTES coating (Figure 3) even though the FF volume concentration is low [50].

Figure 8b,c show that additional well-ordered layers form a three-dimensional structure above this close-packed wetting layer with characteristics (i.e., thicknesses, SLDs, and magnetizations) that depend upon the magnitude of the magnetic field. As schematically depicted in Figure 9b, a double NP layer (# 4), consisting primarily of tilted dimers, forms in a low field of 6 mT. This layer splits into two distinguishable NP layers (# 4a and # 4b in Figure 9c) with magnetizations pointing in different directions in an in-plane magnetic field of 100 mT. This ability to control interface width and roughness can be useful for NP layers in that magnetic hysteresis was found to depend on interface roughness in multilayer films [77]. The field-dependent response of the dilute NP layer (# 5 in Figure 8b,c) adjacent to the free FF was proven to be the primary determinant of the structural and magnetic configurations of the underlying NP layers. Specifically, magnetic frustration can arise because the NPs in layer # 5 are free to rotate in a magnetic field, whereas the NPs in layer # 2 are physically bound to the APTES-coated substrate. By carefully balancing the competing energetics in this system (including the Zeeman, dipolar and steric interactions), it is clearly possible to construct a controllable suite of complex, multilayer NP structures.

### 5.3. Self-Assembly on a Magnetic Surface

Stable NP layers were also observed to self assemble on a 40 nm thick Tb${}_{15}$Co${}_{85}$ film with a domain pattern having an out-of-plane magnetization. The series of iron oxide FFs for this study [52] contained monodispersed, spherical, single-domain NPs of different sizes coated with a monolayer of an ester (N-Hydroxysuccinimide), which is stable in the water solvent and also allows for a chemical interaction that facilitates attachment to a functionalized surface. The nominal concentration of each FF was 0.15 vol% Fe${}_{3}$O${}_{4}$ or 8 mg/mL.

Figure 10a shows the polarized neutron reflectivity for one of the FFs with 22.2 nm diameter NPs in a small guide field. The nuclear SLD profile extracted from fits to the data (solid lines) is plotted in Figure 10b as a function of the distance from the silicon surface.

In contrast to the systems described above in Section 5.1 and Section 5.2, the three components of the NP wetting layer # 1 (i.e., cores and shells below and above) forming on the Tb${}_{15}$Co${}_{85}$ surface (near z= 50 nm) cannot be individually resolved within the trilayer stack, suggesting greater in-plane disorder. Instead, layers # 1 and # 2 describe complete, separate NP layers which are both close-packed, though layer # 2 contains more water. It is notable that these layers remain mostly intact after rinsing the substrate. This investigation opens the possibility of engineering self-assembled NP structures that do not require specialized substrate treatments and can be controlled simply with a magnetic field.

## 6. Summary

The study of magnetic nanoparticle interactions with surfaces is interesting, not only as a testbed for theories of magnetic interactions, but also for a rapidly growing list of technological applications [7,8,9,10,11,12,13,14,15,16,17,18,19,20,21,22,78,79]. These applications include high density magnetic recording media, novel approaches to nanofabrication and medical therapies. Recent studies of magnetic nanoparticles emphasize the importance of characterizing nanoparticle/substrate interface for facilitation of the self-assembly process [80]. NR and PNR are uniquely suited to characterize these structures due to their nanometer scale resolution, isotopic control of contrast and ability to quantitatively probe the structure and vector magnetism as a function of depth into the sample. This article has provided detailed examples of different types of physical interactions among magnetic nanoparticles and their interactions with the surface, some practical aspects of performing neutron reflectivity experiments, and the resultant characteristics revealed for NP self-assemblies prepared under different conditions. The representative selection of the early NR studies reviewed here demonstrated the tendency of magnetic NPs in FFs to order into close-packed layers on nearby surfaces [53,59,60,61,62,63,64,65,66,67]. The three studies that are described in more detail [49,50,52] further highlight the utility of PNR for in-depth insights into the complex nanoscaled structures formed by the interactions of different NPs in different concentrations and with different surfaces.

The first example used a high concentration of slightly non-spherical nanoparticles interacting with a thin silicon oxide surface. By comparing the measured SLDs of the three sub-layers that make up the first monolayer of attached particles, it is demonstrated that this wetting layer can be described as close-packed with excess oleic acid filling the interstices between it and the substrate. Along with two that are less dense, more disordered layers form on top. Upon applying an in-plane 11 mT magnetic field, the nanoparticles in the wetting layer reorient with the long axis parallel to surface, and the disorder in the two outer layers increases.

The second example investigated the ability of a surface to form a wetting layer from a dilute solution of nanoparticles. Here an APTES layer was applied to the silicon surface to increase the interaction strength with the nanoparticles coated with both oleic and carboxylic acids. Here too, a dense, close-packed wetting layer formed, increasing the concentration of nanoparticles relative to the solution by a factor of 160 times. Because of the formation of dimers, the particles beyond this layer formed a more intermixed double layer, but upon increasing the in-plane magnetic field from 6 mT to 100 mT these dimers rotated into the plane of the substrate and the double layer differentiated into two distinct layers with different particle densities.

The third example demonstrated how a magnetic coating can also be used to stabilize a close-packed wetting layer (although it has greater disorder and the ligand shell sublayers are subsequently not distinct) and an additional layer with lower particle density. This result shows potential for devices that make use of the ability to add or subtract the layer of nanoparticles by controlling the magnetization of the surface. This approach can be expanded upon to use the presence of the wetting layer as a fundamental research tool to probe magnetic interaction strength.

Together, these examples demonstrate that a robust close-packed wetting layer can be produced by a variety of surface interactions, both chemical and magnetic, and that its properties can be controlled by several parameters such as particle size and symmetry. The structure of the subsequent layer varies to a greater extent as a function of solution properties and surface interaction characteristics. Finally, an applied magnetic field can be used to control the structures of these layers. These examples, and those of the literature, show how the combination of quantitative depth profiles with nanometer resolution provided by neutron reflectivity and related techniques, with the wide range of choices for surface modifications, nanoparticle properties and solution content, can serve as a fundamental research probe and inspire many practical applications through the control of the surface properties.

## Figures and Tables

**Figure 1 nanomaterials-10-01231-f001:**
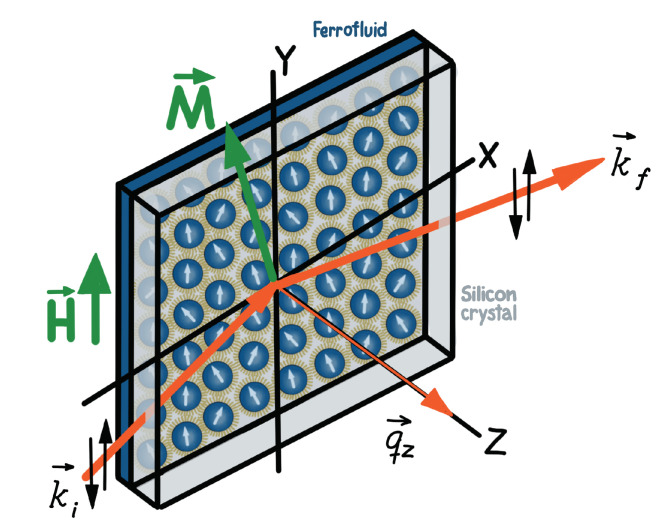
Schematic of the polarized neutron reflectivity measurement configuration that depicts the nanoparticles in the sample cell, along with the spin directions of the incident and scattered neutron beams ($k_{i},k_{f}$). The wave vector $q_{z}$ denotes the momentum transfer perpendicular to the sample plane.

**Figure 2 nanomaterials-10-01231-f002:**
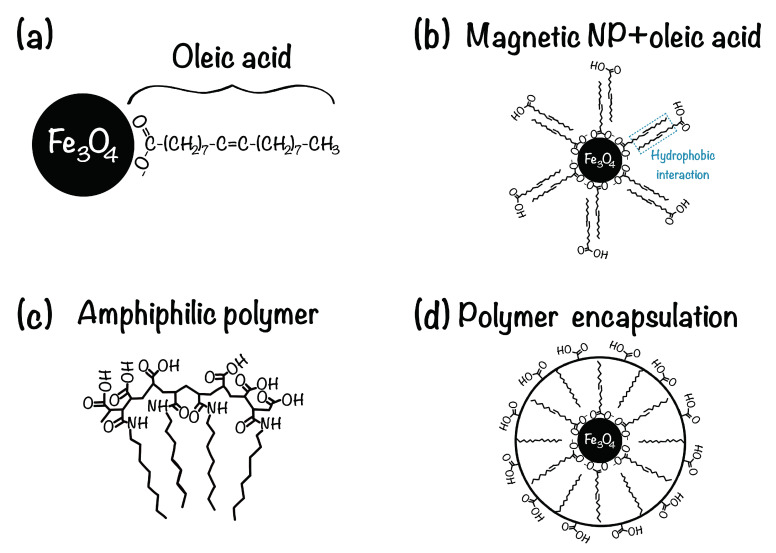
Schematic drawing of an oleic acid coated magnetite NP dispersible in organic solvents (**a**). Water solubility is achieved by addition of surplus oleic acid resulting in a double layer-oleic acid coated NPs (**b**), and by using an amphiphilic polymer (**c**) that encapsulates the NP and native surface ligands (**d**), thereby providing functional molecules capable of secondary conjugation. Figure adapted from reference [73].

**Figure 3 nanomaterials-10-01231-f003:**
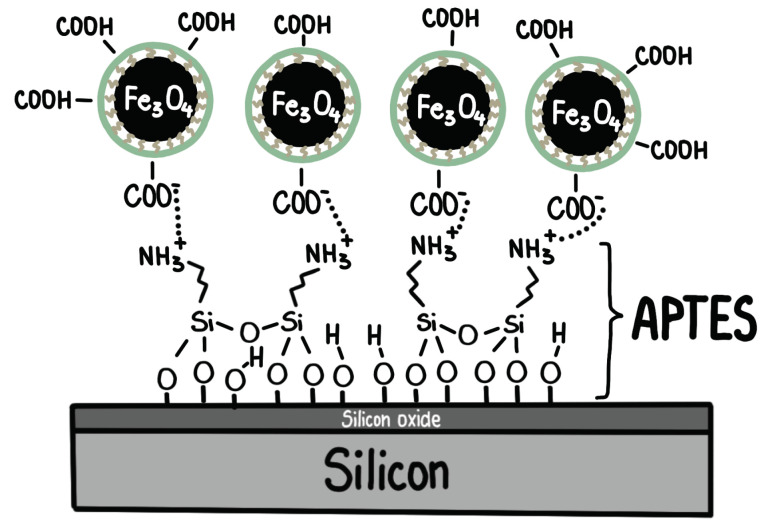
Diagram showing the amine bond between the silane group of the APTES coating on the substrate and the carboxylic group of the ligands in the NPs’ shells [50].

**Figure 4 nanomaterials-10-01231-f004:**
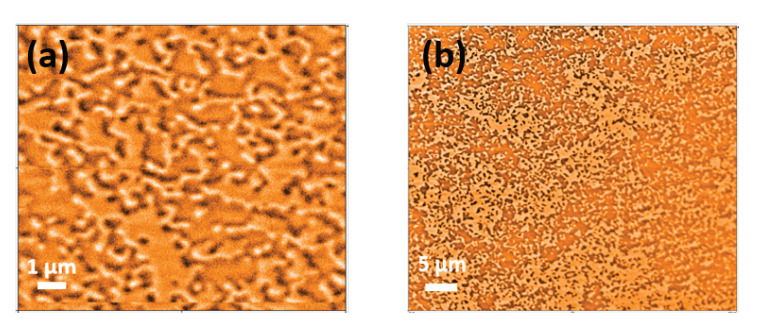
(**a**,**b**) MFM images of the magnetic domain structures of a 40 nm thick Tb${}_{15}$Co${}_{85}$ template grown without an external field. The magnetization points in or out of the sample plane in the dark and light areas, respectively.

**Figure 5 nanomaterials-10-01231-f005:**
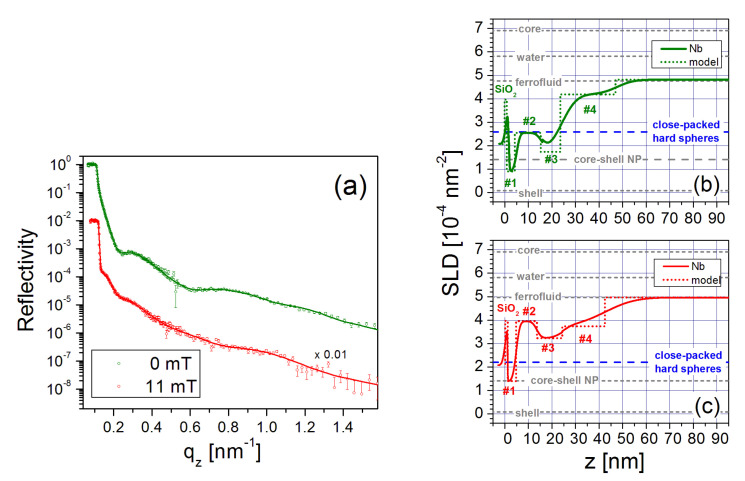
(**a**) NR data and fits (solid lines) for iron oxide nanoparticles on a hydrophilic Si/SiO${}_{2}$ surface in fields of 0 mT and 11 mT. The latter data (red) are scaled by 0.01 for clarity. (**b**,**c**) Profiles of the nuclear SLDs as a function of distance *z* from the Si surface obtained from the model fits to the 0 mT and 11 mT NR data, respectively. [49]

**Figure 6 nanomaterials-10-01231-f006:**
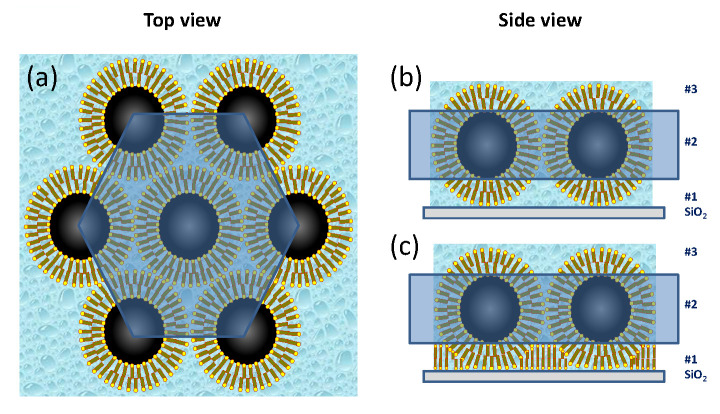
(**a**) Top view of the close-packed arrangement of NPs in a wetting layer with six-fold symmetry, and side views of models for (**b**) close-packed NPs that are intact and (**c**) that have an additional oleic acid layer that wets the substrate and intercalates into the lower NP shell layer [49].

**Figure 7 nanomaterials-10-01231-f007:**
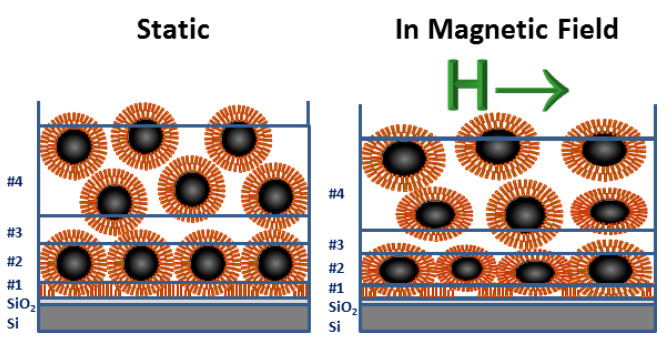
Schematics showing the layers that form in static and magnetic fields for a FF in direct contact with a SiO${}_{2}$ hydrophilic surface. As detailed in the text, layer # 1 is primarily composed of ligands, whereas layer # 2 mostly has close-packed NP cores. Layer # 3 is similar to # 1 but with more evidence of disorder. The NPs are assumed to be spherical in the static model (left). The magnetic field model (right) depicts elongated NPs with long axes that align parallel to the in-plane magnetic field [49].

**Figure 8 nanomaterials-10-01231-f008:**
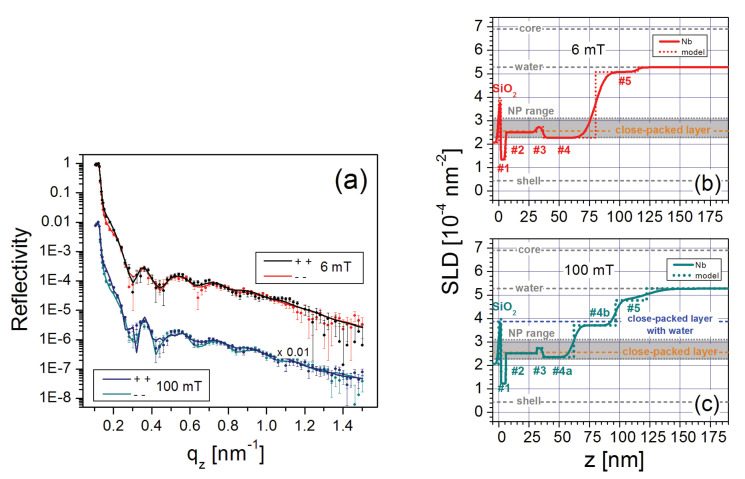
(**a**) PNR data and fits for a FF with monodispersed, spherical NPs on APTES taken in magnetic fields of 6 mT and 100 mT. The data and fits for the 100 mT measurements were scaled by 0.01 for clarity. (**b**,**c**) Profiles of the nuclear SLDs as a function of distance *z* from the Si surface obtained from the model fits to the 6 mT and 100 mT PNR data, respectively [50].

**Figure 9 nanomaterials-10-01231-f009:**
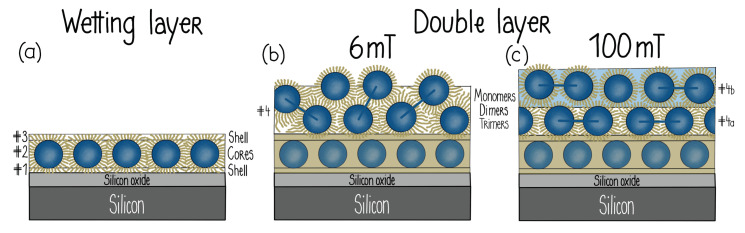
Schematic drawings of NP ordering on an APTES-coated substrate. (**a**) NP wetting layer (# 1–3), (**b**) double layer of tilted NP dimers above the wetting layer in a 6 mT field and (**c**) two distinct NP layers that form above the wetting layer in a 100 mT field [50].

**Figure 10 nanomaterials-10-01231-f010:**
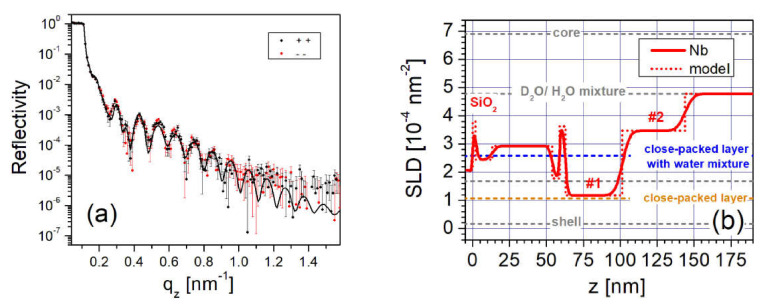
(**a**) PNR data and fit (solid line) for a FF with iron oxide NPs with a diameter of 22.2 nm in contact with a 40 nm Tb${}_{15}$Co${}_{85}$ film in a small in-plane guide field (approximately 1 mT). (**b**) Nuclear SLD profile as a function of distance *z* from the Si surface obtained from model fits to the PNR data. The Tb${}_{15}$Co${}_{85}$ surface is near z= 50 nm. The first and second NP layers (#1 and #2) are close-packed.

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
