# Peer review of "Self-Assembly of Magnetic Nanoparticles in Ferrofluids on Different Templates Investigated by Neutron Reflectometry"

_nanomaterials, 2020, doi:10.3390/nano10061231_

Round 1

Reviewer 1 Report

The manuscript by Katharina Theis-Bröhl and colleagues presents a review of the use of neutron reflectometry as a unique tool to study the properties of self-assembled layers of magnetic nanoparticles in ferrofluids.

Magnetic nanoparticles have been studied intensively in the past two decades, initially because they allow to probe a particular size regime in magnetism that was previously not easily accessible. This has led to much deeper insights into magnetism in confined geometries, at surfaces, in frustrated configurations etc. Together with the progressing insight into the fundamental nature of magnetism at the nanometer scale, real-life applications have emerged and still more are expected, e.g. in magnetic storage technologies, in imaging, in medicat therapies like hyperthermia and so on. It is therefore a good moment for a comprehensive review of the main properties of that class of nanoparticles and to put that in the specific light of the unique probing of such layers by neutron reflectometry.

The selection of examples presented in this review paper is excellent, because they represent the breadth and width of this research field and they highlight the particular information that can be gleaned from neutron reflectometry, so the paper stays very true to what it promises in the title.

My main remark about this manuscript is the following: I would suggest to the authors to expand somewhat more about the technique of neutron reflectometry. Apart from a short piece of text and the related figure 1, not much background is given about the technique as such. Obviously there are good references to literature, but I think that the value of this manuscript would be enhanced it it would be a bit more of a self-containing paper, especially for those who are new or remote to the topic. For instance, section 2 of the paper now opnes with statements about the scattering length density and this is then also what is shown in later figures, but I think that the meaning and relevance of this quantity should be better introduced in the manuscript. I strongly believe that adding some more details on the technique itself will add to the general accessibility of the manuscript.

Apart from the suggestion above, I think this is a very nice and well-written review. Therefore I recommend its publication after the authors have considered the suggestions made above.

Author Response

Dear Referee,

We have revised our manuscript in accordance with the referees’ comments.  Please find a summary of changes made and detailed responses to each of the referee comments i the file attached.  We would like to thank the referees for their careful reading of our manuscript and for their constructive feedback that has helped us to strengthen our manuscript. We now believe that our manuscript is suitable for inclusion in the special issue of Nanomaterials. 

Sincerely,

Dr. Katharina Theis-Broehl

Reviewer 2 Report

The manuscript by Theis-Bröhl et al. aims to provide and overview about the use of neutron reflectivity for studying the depth profile of nanoparticle adlayers at different substrates due to particle adsorption from ferrofluids. 

The work summarises high quality results that are important i the specific filed an I recommend it for publication, but only after major revision as several important issues have to be addressed. These are given in detail below.

  • Figure 2a is quite disturbing as it suggest that there is a coordination between the double bond (or CH2 group) and the CH3 group of the oleic acid, whereas the interaction is dispersive in nature and acts between the alkyl chains if the solvent is appropriate. What about the dissociated carboxil group? Where is the charge? How are both oxigens coordinating (and what)?
  • Also in Figure 2b, the concept of the simultaneous use of the oleic acid and the polymer cannot be easily followed. In the cited reference it says that a monolayer oleic acid is combined with a monolayer of the polymer. Is a mixed monolayer of a bilayer structure? Please try to improve the whole scheme accordingly, so the meaning of the figures and the related textual description becomes more precise.
  • As far as I can judge, Ref. 51 is the only paper in the results section that contains work from outside the group. Its relevance for the particle adsorption study is not clear as Ref. 51 is dealing with the substrate properties. 
  • At this point it is also not clear how the Ref. 52 should be interpreted. Section 5.3 "reviews" these results, but these are not published yet and hence in this form inappropriate. If this is supposed to be a hybrid review also containing new results more details had be given.
  • The term “wetting layer” is very misguiding as it suggest that the formation of the adlayer(s) at the substrate is dictated by wetting. It might be used in the specific field, but in general this term obviously describes how pure liquids wet a substrate. In the present case - as this is an interfacial particle accumulation, not a wetting process - the term adsorption is more correct, so please use this instead.

Summary section:

  • Opposed to what is claimed in the Summary section ("This review has describe the basics of the physics of interactions among magnetic nanoparticles and their interactions with the surface"), details about the physics behind the interaction is missing from the manuscript. Please amend/modify the text appropriately. 
  • The cited 3 studies are actually only 2, as the third one (Ref. 52) cannot be evaluated. If the third one refers to the not-yet-published results, please amend the manuscript to give more details. 
  • The different NP concentrations are specified only in one case (Section 5.1 - 4.6 vol%), otherwise "dilute" (Section 5.2) is used or not given (Section 5.3). These had to be included from the original publications in the main text as well (especially to support the 160x concentration for the APTES modified substrates).

Author Response

(The authors gave the same response as above.)

Round 2

Reviewer 2 Report

The authors provided detailed, point-by-point answers to the comments and modified the manuscript accordingly.

Author Response

We thank the referee for this positive feedback.